Comparative genomic analysis of a new tellurite-resistant Psychrobacter strain isolated from the Antarctic Peninsula

Muñoz-Villagrán Claudia Melissa 1 2
Mendez Katterinne N. 3
Cornejo Fabian 1
Figueroa Maximiliano 1
Undabarrena Agustina 4
Morales Eduardo Hugo 1
Arenas-Salinas Mauricio 5
Arenas Felipe Alejandro 1
Castro-Nallar Eduardo eduardo.castro@unab.cl castronallar@gmail.com 3
Vásquez Claudio Christian claudio.vasquez@usach.cl 1
1 Laboratorio de Microbiología Molecular, Departamento de Biología, Universidad de Santiago de Chile , Santiago , Chile
2 Departamento de Ciencias Básicas, Facultad de Ciencia, Universidad Santo Tomas Sede Santiago , Santiago , Chile
3 Center for Bioinformatics and Integrative Biology, Facultad de Ciencias Biológicas, Universidad Andrés Bello , Santiago , Chile
4 Laboratorio de Microbiología Molecular y Biotecnología Ambiental, Departamento de Química & Centro de Biotecnología Daniel Alkalay Lowitt, Universidad Técnica Federico Santa María , Valparaíso , Chile
5 Centro de Bioinformática y Simulación Molecular, Universidad de Talca , Talca , Chile
Vishnivetskaya Tatiana
Electronic publication date: 2018 Feb 19
Publication date: 2018
Volume: 6
Electronic Location ID: e4402
Received 2017 Oct 12; Accepted 2018 Feb 1
Copyright: ©2018 Muñoz-Villagrán et al.
Copyright year: 2018
Copyright holder: Muñoz-Villagrán et al.
License: This is an open access article distributed under the terms of the Creative Commons Attribution License, which permits unrestricted use, distribution, reproduction and adaptation in any medium and for any purpose provided that it is properly attributed. For attribution, the original author(s), title, publication source (PeerJ) and either DOI or URL of the article must be cited.
License URL: https://creativecommons.org/licenses/by/4.0/

Keywords: Ter genes, Antarctica, Extremophiles, Tellurite resistance, Phylogenomics

Funding: Fondecyt (Fondo Nacional de Ciencia y Tecnología) 1130362 11140334 3150004 1160051 Conicyt (Comisión Nacional de Investigación Científica y Tecnológica) 21120154 INACH (Instituto Antártico Chileno) DG_03-13 Fondo de Proyectos para Investigadores Iniciales—Universidad de Talca Fondecyt de Iniciación en Investigación 11160905 This work was supported by Fondecyt (Fondo Nacional de Ciencia y Tecnología) grants (1130362, 11140334, 3150004, 1160051), Conicyt (Comisión Nacional de Investigación Científica y Tecnológica) doctoral fellowship (21120154), INACH (Instituto Antártico Chileno) fellowship (DG_03-13). Mauricio Arenas-Salinas was funded by Fondo de Proyectos para Investigadores Iniciales—Universidad de Talca. Eduardo Castro-Nallar was funded by Fondecyt de Iniciación en Investigación grant (11160905). The funders had no role in study design, data collection and analysis, decision to publish, or preparation of the manuscript.

==============================
The Psychrobacter genus is a cosmopolitan and diverse group of aerobic, cold-adapted, Gram-negative bacteria exhibiting biotechnological potential for low-temperature applications including bioremediation. Here, we present the draft genome sequence of a bacterium from the Psychrobacter genus isolated from a sediment sample from King George Island, Antarctica (3,490,622 bp; 18 scaffolds; G + C = 42.76%). Using phylogenetic analysis, biochemical properties and scanning electron microscopy the bacterium was identified as Psychrobacter glacincola BNF20, making it the first genome sequence reported for this species. P. glacincola BNF20 showed high tellurite (MIC 2.3 mM) and chromate (MIC 6.0 mM) resistance, respectively. Genome-wide nucleotide identity comparisons revealed that P. glacincola BNF20 is highly similar (>90%) to other uncharacterized Psychrobacter spp. such as JCM18903, JCM18902, and P11F6. Bayesian multi-locus phylogenetic analysis showed that P. glacincola BNF20 belongs to a polyphyletic clade with other bacteria isolated from polar regions. A high number of genes related to metal(loid) resistance were found, including tellurite resistance genetic determinants located in two contigs: Contig LIQB01000002.1 exhibited five ter genes, each showing putative promoter sequences (terACDEZ), whereas contig LIQB1000003.2 showed a variant of the terZ gene. Finally, investigating the presence and taxonomic distribution of ter genes in the NCBI’s RefSeq bacterial database (5,398 genomes, as January 2017), revealed that 2,623 (48.59%) genomes showed at least one ter gene. At the family level, most (68.7%) genomes harbored one ter gene and 15.6% exhibited five (including P. glacincola BNF20). Overall, our results highlight the diverse nature (genetic and geographic diversity) of the Psychrobacter genus, provide insights into potential mechanisms of metal resistance, and exemplify the benefits of sampling remote locations for prospecting new molecular determinants.

Introduction

The bacterial genus Psychrobacter was first described by Juni & Heym (1986) and includes a group of non-motile, oxidase positive, psychrotolerant, Gram-negative rods or coccobacilli isolated from animals and processed foods (Bozal et al., 2003). Bacteria from the Psychrobacter genus have also been isolated from natural environments such as Antarctic ornithogenic soils, sea ice, deep-sea, and sea water from the Pacific Ocean and other locations (Bowman, Nichols & McMeekin, 1997; Romanenko et al., 2002). Antarctic isolates belonging to the genus Psychrobacter have been described and identified as P. inmobilis, P. glacincola, P. luti and P. fozzi (Bozal et al., 2003).

The Antarctic territory is the coldest and driest environment on the planet and is exposed to high levels of UV radiation, which favors the production of intracellular Reactive Oxygen Species (ROS) (D’Amico et al., 2006; Potts, 1994). Consequently, microbial communities residing in the Antarctica are unique and some possess diverse strategies to cope with the deleterious effects of ROS and other extreme conditions. In agreement, several microorganisms resistant to antibiotics and other toxicants including tellurite have been isolated from this environment (Arenas et al., 2014; De Souza et al., 2006; Lo Giudice et al., 2013). Many of these organisms can grow at low temperatures and tolerate/resist different compounds, making them ideal candidates for biotechnological applications such as the production of polyunsaturated fatty acids, bioremediation, or as source of industrially useful enzymes (e.g., proteases, lipases) (Brenchley, 1996; Denner et al., 2001).

Tellurite is extremely harmful to most microorganisms, and its toxicity has been associated with the establishment of an oxidative stress status, including ROS generation (Chasteen et al., 2009; Pérez et al., 2007). These ROS are produced as a byproduct of tellurite reduction to its elemental form by either enzymatic or non-enzymatic mechanisms, as visualized by the accumulation of black deposits near the bacterial membrane (Amoozegar et al., 2008; Chasteen & Bentley, 2003; Taylor et al., 1988). Other metal(oid) resistance mechanisms commonly found in bacteria include global cellular responses, cell grouping, uptake control and oxidative stress response, among others (Lemire, Harrison & Turner, 2013).

Some of the genes implicated in tellurite resistance include trgAB from Rhodobacter sphaeroides (unknown function, encoding likely membrane-associated proteins) (O’Gara, Gomelsky & Kaplan, 1997), tmp from Pseudomonas syringae (encoding a thiopurine methyltransferase, involved in tellurium alkylation) (Prigent-Combaret et al., 2012), lpdA from Aeromonas caviae ST (encoding dihydrolipoamide dehydrogenase, involved in tellurite reduction) (Castro et al., 2008) and gor from Pseudomonas sp. (encoding glutathione reductase, involved in tellurite reduction) (Arenas et al., 2016; Pugin et al., 2014). Tellurite resistance genes were also identified in the ter operon (terZABCDEF) from Escherichia coli and other pathogenic species (Taylor, 1999; Whelan, Sherburne & Taylor, 1997).

Interestingly, the ter operon is not only associated with tellurite resistance but also with resistance to bacteriophage infections and to antimicrobial compounds like colicins (Whelan, Colleran & Taylor, 1995). The mechanism of action of the proteins encoded by the ter operon remains to be elucidated; however, it is known that they form a multi-subunit complex associated with the inner surface of the bacterial membrane (Anantharaman, Iyer & Aravind, 2012). Likewise, only terB, terC, terD and terE have been shown to be directly involved in tellurite resistance (Taylor et al., 2002).

During the Chilean Antarctic expedition ECA-48 in 2012, a bacterium—later identified as P. glacincola BNF20—was isolated and characterized; although P. glacincola BNF20 was highly resistant to tellurite (MIC 2.3 mM), it did not show increased ROS levels or tellurite reductase activity when exposed to the toxicant (Arenas et al., 2014). In this work, we determined for the first time the genome sequence of a member of the P. glacincola species. To gain insight into the potential mechanism(s) of tellurite resistance, we conducted a comparative genomics analysis using available Psychrobacter genome sequences. Specifically, we tested if P. glacincola BNF20 was phylogenetically related to other Psychrobacter Antarctic isolates, and if it harbors ter genes. Finally, the ter gene taxonomic distribution was assessed using a reference database containing over 5,000 bacterial genomes.

Materials and Methods

Strain isolation and culture conditions

P. glacincola BNF20 was isolated from a sediment sample collected at King George Island, Antarctica (S62°11′37.6″; W58°56′14.9″) during the ECA-48 Chilean Antarctic Expedition (January 2012). Bacteria were grown at 25 °C as described previously (Arenas et al., 2014) in Lysogenic Broth (LB) medium (Sambrook & Russell, 2001) supplemented with tellurite (200 µg/ml). Strains were identified by sequencing the 16S rRNA gene (accession MF806171) and determining the fatty acid profile. The 16S rRNA gene was sequenced at Pontificia Universidad Católica de Chile using Sanger sequencing with the primers 8F (5′-AGAGTTTGATCCTGGCTCAG-3′) (Turner et al., 1999) and 1492R (5′-ACGGCTACCTTGTTACGACTT-3′) (Lane, 1991). Fatty acid analyses were carried out at DSMZ, Braunschweig, Germany (Kämpfer & Kroppenstedt, 1996). The strain was deposited at DMSZ (Germany), accession # 102806.

To determine the minimal inhibitory concentration (MIC), bacteria were grown overnight in LB medium with shaking at 25 °C. Subsequently, saturated cultures were diluted 1:100 with fresh medium and grown to OD600 ∼ 0.4–0.5. Then, 10 µl were added to 990 µl of LB medium containing serial dilutions of defined toxicants in 48-well culture plates. The plates were incubated with constant shaking for 24 h at 25 °C. Assayed toxicants included K2TeO3, K2CrO4, CdCl2, ZnCl2, CuSO4, HAuCl4, AgNO3, NiSO4, NaAsO2 and Na2HAsO4.

16S rRNA gene phylogenetic analysis

A phylogenetic tree of P. glacincola BNF20—based on the partial 16S rRNA gene sequence—was constructed with bootstrap values based on 1,000 replications (Felsenstein, 1985). The nearly complete 16S rRNA gene sequence (1,516 nt) was obtained by merging the PCR sequenced amplicons (accession MF806171) and the sequence obtained by whole genome shotgun sequencing (accession AMK37_RS07000). Sequence alignments, assembly and comparisons, along with best model calculation and construction of the phylogenetic tree were carried out using the MEGA software version 6.0 (Tamura et al., 2013). As outcome, Jukes Cantor model and Pairwise Deletions for gaps treatment was the best fitting model for these sequence data. Nucleotide sequence positions from 16 to 1,535 were considered, according to the E. coli K-12 16S rRNA gene sequence numbering (accession AP012306). Scale bar represents 0.01 substitutions per-nucleotide positions. Moraxella osioensis DSM 6998T (JN175341) was used as outgroup. The following 16S rRNA sequences from Psychrobacter strains were collected from GenBank (accession numbers are given in parentheses): P. glacincola DSM 12194T (AJ312213); P. adeliensis DSM 15333T (AJ539105); P. urotivorans DSM 14009T (AJ609555); P. arcticus DSM 17307T (AY444822); P. cibarius DSM 16327T (AY639871); P. cryohalolentis DSM 17306T (AY660685); P. frigidicola DSM 12411T (AJ609556); P. fozii NF23T (AJ430827); P. inmobilis DSM 7229T (U39399); P. namhaensis DSM 16330T (AY722805); P. aquimaris DSM 16329T (AY722804); P. luti NF11T (AJ430828); P. alimentarius DSM 16065T (AY513645); P. fulvigenes JCM 15525 (AB438958); P. piscatorii JCM 15603 (AB453700); P. jeotgalli JCM 11463T (AF441201); P. arenosus DSM 15389T (AJ609273) and P. okhotskensis JCM 11840 (AB094794).

Preparation of genomic DNA

P. glacincola BNF20 was grown in LB medium at 25 °C for 24 h with constant shaking. DNA was extracted using the Wizard Genomic® DNA Purification Kit (Promega, Madison, WI, USA). The quality and integrity of gDNA was determined by agarose gel (1%) electrophoresis and by determining the 260/280 nm absorbance ratio in a microplate multireader equipment (Tecan Infinite®PRO; Tecan, Männedorf, Switzerland).

Genome sequencing and annotation

The draft genome sequence of P. glacincola BNF20 was determined by a whole-genome shotgun strategy using the Illumina HiSeq 2000 platform with a mate-pair library of 3 kb (Macrogen®). A total of 10.89 million reads were quality filtered and assembled using the A5 pipeline (Tritt et al., 2012). Open reading frame prediction and annotation was carried out using standard operational procedures (Tanenbaum et al., 2010). Gene models were predicted using Glimmer 3.02 (Salzberg et al., 1998) and predicted coding sequences were annotated by comparison with public databases (COG, PFAM, TIGRFAM, UNIPROT, and NR-NCBI). P. glacincola BNF20 predicted proteome completeness was assessed by the presence/absence of bacterial orthologs according to the OrthoDB database using BUSCO (Simão et al., 2015). The circular genome map was assembled from P. glacincola BNF20 GenBank formatted file (NZ_LIQB00000000.1) using the plotMyGBK wrapper script (https://github.com/microgenomics/plotMyGBK); plotMyGBK uses BioPython and the R platform with the packages rSamTools, OmicCircos, and data.table to produce a vector image of a circular map (Cock et al., 2009; R Development Core Team, 2011; Morgan et al., 2016; Hu et al., 2014; https://github.com/Rdatatable/data.table).

Nucleotide sequence accession and culture collection number

Raw sequence data from P. glacincola BNF20 are available online under the BioProject #PRJNA293364, and Gold ID Gp0145575. The genome project has been deposited at GenBank, accession number NZ_LIQB00000000. The strain was deposited at the DSMZ culture collection, ID number DSM 102806.

Psychrobacter genome dataset

A total of 35 Psychrobacter genomes including P. glacincola BNF20 were retrieved from NCBI’s Genome and JGI GOLD databases (as of February 2017), where 10 and 24 genomes were annotated to the species and genus level, respectively.

Phylogenetic relationships and whole-genome nucleotide identity

The average nucleotide identity (ANI) was calculated for the 35-genome dataset using the pyani Python3 module (Pritchard et al., 2016) and the results were visualized using the data.table and pheatmap R packages (https://github.com/Rdatatable/data.table; https://cran.r-project.org/web/packages/pheatmap/index.html). Thirty-one phylogenetic marker genes corresponding to widespread housekeeping genes dnaG, nusA, rplA, rplD, rplK, rplN, rplT, rpsB, rpsI, rpsM, tsf, frr, pgk, rplB, rplE, rplL, rplP, rpmA, rpsC, rpsJ, rpsS infC, pyrG, rplC, rplF, rplM, rplS, rpoB, rpsE, rpsK, and smpB were identified in each Psychrobacter genome (AMPHORA2; Wu & Eisen, 2008). Each gene was translated under standard genetic code to perform a protein-coding-guided multiple nucleotide sequence alignment, using TranslatorX MUSCLE for the multiple sequence alignment (Abascal, Zardoya & Telford, 2010; Edgar, 2004). Alignments were concatenated using the alignment editor tool Seqotron (Fourment & Holmes, 2016) and the best partition scheme and substitution model was evaluated by PartitionFinder2 (Lanfear et al., 2016). Finally, the software MrBayes v3.2 was used for phylogenetic reconstruction (Ronquist et al., 2012), and the resulting tree was plotted and annotated using FigTree v1.4.3 (http://tree.bio.ed.ac.uk/software/figtree/). Phylogenetic tree annotation was based on the geographic location, according to BioSample database information for each Psychrobacter genome.

Search for metal resistance ortholog genes

To identify metal resistance genes, especially ter genes, a bidirectional Blast analysis was performed using the CRB-BLAST method (https://github.com/cboursnell/crb-blast). The BacMet Metal Resistance database (Pal et al., 2014) was used as target and the 35-genome dataset as query. In addition, each genome was re-annotated using the same methodology to identify syntenic genes based on BacMet and Prokka annotation without the bias of different annotation labels as implemented in Prokka v1.12 (Seemann, 2014). Finally, the results were visualized in their genomic context using the in-house script multiGenomicContext (https://github.com/Sanrrone/multiGenomicContext).

Promoter search

To elucidate if ter genes were under the control of one or more promoters, two promoter prediction tools were used on specific contigs where ter genes were found (LIQB01000002.1, position 189803-204267; LIQB01000003.2, position 286133-301767): PromPredict algorithm and the online program BPROM (Rangannan & Bansal, 2010; http://www.softberry.com/berry.phtml?topic=bprom&group=programs&subgroup=gfindb). Both results were jointly analyzed.

Taxonomic classification of tergenes

To investigate if the presence and number of ter genes was restricted to certain taxonomic levels, we downloaded all the bacterial reference genomes from the NCBI’s RefSeq database (January 2017; ftp://ftp.ncbi.nlm.nih.gov/refseq/release/bacteria/) and performed a bidirectional Blast searches (crb-blast) against the protein sequences of all ter genes in the BactMet database. Then, the in-house script fetchMyLineage (https://github.com/Sanrrone/fetchMyLineage) was employed to obtain the complete lineage of each bacterial genome with at least one ter gene match. The results were finally visualized using the R packages: ggplot2, RColorBrewer, devtools, ggjoy purr andreshape2 packages (R Development Core Team, 2011; http://ggplot2.org; https://cran.r-project.org/web/packages/RColorBrewer/index.html; https://github.com/hadley/devtools; https://cran.r-project.org/web/packages/ggjoy/index.html; Wickham, 2007).

Results

A new Psychrobacter species from the Chilean Antarctic territory

P. glacincola BNF20 was isolated from Antarctic sediments and is a Gram-negative, non-motile, aerobic, oxidase positive, rod-shaped bacterium with an average dimension of 1.66 µm length and 1.09 µm width (Table 1, Fig. 1A). The fatty acid composition was determined at the DSMZ Institute (Germany) and showed that the major fatty acid was cis-9 octadecenoic acid C18:1 ω9c (63.78%). The morphology description and major fatty acid component agrees with previous studies of Antarctic Psychrobacter isolates (Bozal et al., 2003). Initially, BNF20 was erroneously identified as P. inmobilis, based on a partial 16S rRNA gene sequence (Arenas et al., 2014). However, re-sequencing and a phylogenetic analysis of the partial 16S rRNA gene revealed that it is related to the P. glacincola species, family Pseudomonadaceae from the Gammaproteobacteria class (Fig. 1B). Altogether, morphology (electron microscopy), biochemical properties, partial (Sanger) and full length (NGS) 16 S rRNA gene sequence analysis, and fatty acid composition suggest that isolate BNF20 is member of the P. glacincola species.

Table 1 Classification and general features of P. glacincola BNF20.

MIGS ID	Property	Term	
	Classification	Domain: Bacteria	
		Phylum: Proteobacteria	
		Class: Gammaproteobacteria	
		Order: Pseudomonadales	
		Family: Moraxellaceae	
		Genus: Pychrobacter	
		Species: Psychrobacter glacincola	
		(Type) strain: BNF20	
	Gram stain	Negative	
	Cell shape	Rod	
	Motility	Non-Motile	
	Sporulation	Non-sporulating	
	Temperature range	Psychrotolerant	
	Optimum temperature	25 °C	
	pH range; Optimum	Not tested; 7.4	
	Carbon source	Citrate, acetate, pyruvate	
MIGS-6	Habitat	Antarctic sediment	
MIGS-6.3	Salinity	0–10% NaCl (w/v)	
MIGS-22	Oxygen requirement	Aerobic	
MIGS-15	Biotic relationship	Free-living	
MIGS-14	Pathogenicity	Potentially pathogenic	
MIGS-4	Geographic location	King George Island, Antarctica	
MIGS-5	Sample collection	January, 2012	
MIGS-4.1	Latitude	62°11′S	
MIGS-4.2	Longitude	58°56′W	
MIGS-4.4	Altitude	Not registered	

Figure 1 Phylogenetic, morphological and genomic characteristics of P. glacincola BNF20.

(A) Scanning electron micrograph showing the morphology and dimensions of P. glacincola BNF20. Samples were stained with uranyl acetate (0.5% w/v) and examined using a low-voltage electron microscope (Delong Instruments, LVEM5), with a nominal operating voltage of 5 kV. Bar represents 10 µm. (B) Phylogenetic tree of P. glacincola BNF20 based on the partial 16S rRNA gene sequence (Accession number MF806171). Psychrobacter ingroup was rooted using Moraxella osloensis DSM 6978T as outgroup. (C) Circular map of the 18-scaffold draft genome with coding sequences colored by COG categories. Inner circles represent GC Skew and GC content.

P. glacincola BNF20 tolerates high tellurite and chromate concentrations

Several tests were carried out to determine if BNF20 was resistant to multiple metals. Besides tellurite (used in the initial selection), P. glacincola BNF20 was 4 times more resistant to chromate than the sensitive strain E. coli BW25113 under optimal growth conditions (Table 2). However, P. glacincola BNF20 growth was impaired in the presence of all other metal(loid)s tested, including Cu2+, Cd2+, Hg2+, Zn2+, AuCl41−, Ni2+, AsO42−, AsO31−, and Ag1+.

Table 2 Minimal inhibitory concentrations (mM) of different metal(loid)s for P. glacincola BNF20, and E. coli BW25113 (reference).

Metal	BNF20	BW25113	
TeO32−	2.3	0.004	
Cu2+	3.12	6	
Cd2+	0.062	1	
Hg2+	0.0039	0.01	
Zn2+	0.5	2	
CrO42−	6	1.5	
AuCl41−	0.015	0.16	
Ni2+	1.25	5	
AsO42−	40	80	
AsO21−	5	10	
Ag1+	0.015	0.063	

First draft genome of Psychrobacter glacincola BNF20

Previous studies showed that P. glacincola BNF20 was highly resistant to tellurite (MIC ∼2.3 mM, Arenas et al., 2014). Although tellurite reduction is often accompanied by the formation of black deposits of elemental tellurium in resistant organisms, this phenotype was not observed in P. glacincola BNF20. To further investigate the mechanism(s) of tellurite resistance in P. glacincola BNF20, we sequenced the whole genome in search for genetic determinants implicated in metal(loid) resistance. The assembled genome of P. glacincola BNF20 consisted of 3,490,622 bp, 18 scaffolds, with an average G + C content of 42.76% (Fig. 1C; NCBI Reference Sequence: NZ_LIQB00000000.1). The predicted proteome scored 100% completeness according to the presence of highly conserved ortholog genes in bacteria (BUSCO analysis). A set of 47 tRNA genes and one copy of the rRNA operon were also identified. From a total of 2,968 predicted CDS, 2,872 (96.7%) ORFs matched coding sequences available in public databases, of which 2,515 were assigned (84.7%) or not (352 CDS, 19.31%) to COG categories (NZ_LIQB00000000.1).

P. glacincola BNF20 is evolutionarily divergent from other Antarctic Psychrobacterisolates

The genome sequence of P. glacincola BNF20 was compared to other 34 available genomic sequences by estimating ANI values and performing a multi-locus phylogenetic analysis (Fig. 2). Besides P. glacincola BNF20, the full dataset was composed of 10 named and 24 unnamed Psychrobacter species, respectively. P. glacincola BNF20 exhibited an average nucleotide identity >95% and an alignment fraction of over 80% with 3 isolates designated as Psychrobacter sp. JCM18903 (GCA_000586475.1), Psychrobacter sp. JCM 18902 (GCA_000586455.1) (Kudo et al., 2014) and Psychrobacter sp. P11F6 (GCA_001435295.1) (Moghadam et al., 2016), of which none was isolated from Antarctica. We did not find any genome comparison against BNF20 of >96.5% ANI and >60% alignment fraction, which has been suggested as a “genomic boundary” for bacterial species (Fig. 2A; Varghese et al., 2015). While some of the available genomes come from Antarctic isolates, none of them showed high ANI values (>90%): P. aquaticus (85%; GCA_000471625.1); P. alimentarius (85%; GCA_001606025.1); P. urativorans (85%; GCA_001298525.1); TB15 (84%, GCA_000511655.1), G (86%, GCA_000418305.1); PAMC 21119 (86%, GCA_000247495.2); TB2 (84%, GCA_000508345.1); TB47 (86%, GCA_000511045.1); TB67 (86%, GCA_000511065.1) and AC24 (86%, GCA_000511635.1).

Figure 2 Whole genome nucleotide identity and multi-locus phylogenetic analysis.

(A) Average nucleotide identity (ANI) in the 35-genome Psychrobacter dataset. P. glacincola BNF20 forms a cluster with other three Psychrobacter genomes with an alignment fraction over 80%. (B) Bayesian multi-locus phylogenetic analysis of the genomic sequences from the indicated Psychrobacter members. Taxa are colored by geographic location. Node values correspond to posterior probabilities, and the phylogeny was mid-point rooted.

Supporting our previous results, multi-locus phylogenetic analysis showed that P. glacincola BNF20 is more related to P11F6 (isolated from Tunicate ascidians from the Arctic, Moghadam et al., 2016), JCM 18902 and 18903 (isolated from frozen porpoise Neophocaena phocaenoides, Kudo et al., 2014). Antarctic isolates PAMC21119 and G (from King George Island, Moghadam et al., 2016; Che et al., 2013) belong to a polyphyletic group and do not form a monophyletic clade with P. glacincola BNF20, highlighting the heterogeneous nature of the Psychrobacter genus (Fig. 2B). All nodes of the phylogeny were well supported (posterior probability >  0.99).

P. glacincola BNF20 encodes multiple metal resistance determinants

As P. glacincola BNF20 was isolated from King George Island sediments, a place where heavy metal contamination has not been previously reported, we searched for genes known to be involved in metal resistance that could explain the observed tellurite and chromate resistance of strain BNF20 (BacMet database; Pal et al., 2014). Type and gene copy number distribution was not uniform in the 35-genome Psychrobacter dataset (Table S3).

Specifically, ∼100 genes possibly conferring metal resistance were identified in the genome of P. glacincola BNF20, of which some are related to chromate resistance, including chrL (BAC0361; regulatory protein, involved in chromate resistance), chrR (BAC0538; chromate reductase), mdrL/yfmO (BAC0209; multidrug efflux protein yfmO) and ruvB (BAC0355; ATP-dependent DNA helicase), and some to tellurite resistance—the so-called ter genes (Whelan & Colleran, 1992), including terA (BAC0386), terC (BAC0388), terD (BAC0389), terE (BAC0390) and terZ (BAC0392) (Fig. 3, Table S3). Two other genes apparently involved in tellurite resistance, ruvB (BAC0355; ATP-dependent DNA helicase) and pitA (BAC0312; low-affinity inorganic phosphate transporter 1), were also identified.

Figure 3 Genomic context of the ter genes harbored by P. glacincola BNF20.

(A) Context of the gene cluster located at nucleotide (nt) positions 189,803-204,267 of the Contig LIQB1000002.1. (B) Contig LIQB01000003.2, located at nt positions 286,133–301,767 exhibits an extra copy of the terZ gene.

Organization of ter genes in P. glacincola BNF20

Given that (i) tellurite is by far more toxic for bacteria than other metals (Taylor, 1999) and (ii) it is scarce in the Earth’s crust (Turner, Borghese & Zannoni, 2012), finding tellurite resistance determinants in P. glacincola BNF20 was somewhat unexpected. Since to date the presence of ter genes in Antarctic microorganisms has not been reported, we focused the following analyses our study on them.

The ter genes were originally described as part of an E. coli operon exhibiting the terZABCDE structure (Taylor et al., 2002). P. glacincola BNF20 harbors terA, terZ, terE, terC and terD orthologs, but not terB (Fig. 3A); terA shows the opposite transcriptional orientation than the rest of the ter genes, while terZ is duplicated and is contained in different contigs (Fig. 3B). In addition, the expression of all ter genes in P. glacincola BNF20 seems to be regulated by individual promoters (PromPredict and BPROM analyses), suggesting that they are organized as a gene cluster rather than as an operon.

Three members of the Psychrobacter genus contained one ter gene (P. phenylpyruvicus (terZ, GCA_000685805.1), P. lutiphocae (terZ, GCA_000382145.1) and P. sp. ENNN9 III (terD, GCA001462175.1)), while the rest had different combinations of them (Fig. S1).

In P. glacincola BNF20, the context of the ter gene cluster is similar to other isolates like Psychrobacter sp. JC18902 (GCA_00058655.1), Psychrobacter sp. G (GCA_000418305.1), Psychrobacter sp. TB67 (GCA_000511065.1), Psychrobacter sp. AC24 (GCA_000511635.1), Psychrobacter sp. TB47 (GCA_00051045.1) and P. arcticus 273-4 (GCA_000012305.1). Interestingly, in all analyzed Psychrobacter genomes the ter gene cluster also contains a gene encoding a protein of the TIGR00266 family (unknown function, Fig. S1).

ter genes are distributed over several bacterial Phyla

To determine the frequency of ter genes in known bacterial genomes, their taxonomic distribution was evaluated. In general, ter genes are more commonly found in Gram-positive than in Gram-negative bacteria. Using NCBI’s RefSeq database (>5,000 genomes; accessed January 2017), we found that 48.59% of them contained ter genes (26 out of 30 bacterial Phyla). While, at the genus level, most genomes had one ter gene (67.95%) (Table S4, Fig. 4), others harbor two (2.31%), three (0.69%), four (5.24%), six (4.61%) or seven (1.15%) ter genes. Interestingly, the second most abundant combination of ter genes in genomes was five (18.04%), which could suggest evolutionary constrains.

Figure 4 Number of ter genes in Bacterial families.

Distribution of ter genes present in the indicated phyla. Only taxonomic classifications (Phylum and Family) with at least three bacterial genomes encoding at least one ter gene are shown.

At the phylum level most Proteobacteria contain one ter gene, with a few exceptions showing up to 7, including Yersiniacee, Morganellaceae, Enterobacteriaceae and Erwiniaceae. A similar pattern is observed in other Phyla, except for Firmicutes where genomes exhibit a defined array of ter genes (Fig. 4). Interestingly, while members belonging to the best represented family in RefSeq, i.e., Streptomycetaceae (149 genomes) exhibit five or six ter genes, in other well-represented families such as Flavobactericidae only 26 out of 114 genomes exhibit five ter genes (23%). Within the Moraxelaceae family, nine out of 45 genomes show five ter genes (20%, including BNF20), which agrees with the complete family database distribution (∼18% with 5 ter genes).

Discussion

Here we show for the first time the genome sequence of a P. glacincola species isolated from Antarctica, which can tolerate high concentrations of tellurite and chromate. P.  glacincola BNF20 showed to be 4- and 500-fold more resistance to chromate and the tellurium oxyanion tellurite than E. coli BW25113 (Table 2).

Previous studies showed that defined toxicants can trigger common responses or repair mechanisms (Miranda et al., 2005), suggesting that tellurite and chromate resistance could be related. Besides tellurite and chromate, P. glacincola BNF20 genome encodes resistance determinants associated to a number of other heavy metal(loid)s such as arsenic, cadmium, copper and mercury (Table S3). Interestingly, tellurite resistance in P. glacincola BNF20 did not correlate with a strong tellurite reduction, as previously reported (Arenas et al., 2014), which prompted us to search for genes associated with tellurite resistance in its genome. Identifying these genetic resistance determinants could be useful as the Psychrobacter genus has been proposed as good candidate for biotechnological applications including bioremediation (Lasa & Romalde, 2017).

Members of the Psychrobacter genus are versatile and have been isolated from different places with low temperatures—including Antarctica—as well as from some animal hosts including skin, fish gills and guts and human blood, among others (Bowman, Nichols & McMeekin, 1997; Bozal et al., 2003; Romanenko et al., 2002). However, isolates from similar environments show high genomic variability, as evidenced by ANI analysis (Table S2). A multi-locus phylogenetic analysis revealed that Antarctic Psychrobacter isolates do not form a monophyletic group (Fig. 2). In this context, the presence of ter genes is correlated to some extent with their genomic structure. In fact, higher ANI values reflected a more similar ter gene organization. Thus, P. glacincola BNF20 exhibited a very close ter gene organization with the three closest members Psychrobacter sp. P11F6, JCM18902 and JCM18903 (Fig. 2, Fig. S1).

Psychrophilic and psychrotolerant microorganisms require several genes to increase their phenotypic flexibility to survive in extreme environments such as cold habitats. Thus, in addition to genes associated with cold shock proteins, membrane fluidity, among others, the presence of metal(loid) resistance genes seems to favor their adaptation (Dziewit & Bartosik, 2014; Rodríguez-Rojas et al., 2016). This is also the case of P.  glacincola BNF20, which harbors over 100 putative metal resistance genes (Table S2). In principle and even though this high number of genes predicted bacterial resistance to a number of metal(loid)s, MIC determinations showed that P. glacincola BNF20 was only resistant to chromate and tellurite (MIC 6 and 2.3 mM, respectively). Chromate resistance genes included chrI (regulatory protein of Ralstonia metallidurans CH34; Juhnke et al., 2002), chrR (encoding a chromate reductase; Park et al., 2000), mdrL/yfmO (multidrug efflux transporter in Listeria monocytogenes; Mata, Baquero & Pérez-Díaz, 2000) and ruvB, encoding a DNA helicase involved in both chromate and tellurite resistance in P. aeruginosa PAO1 (Miranda et al., 2005). Genes related to tellurite resistance identified in P. glacincola BNF20 included the phosphate transporter pitA (Elías et al., 2012) and a cluster of ter genes (Whelan, Colleran & Taylor, 1995), composed of terA, terZ, terC, terD and terE, which exhibit a different organization as compared to other ter gene clusters previously described (Fig. 3). Although ter refers to tellurite resistance, the same genes participate in resistance to phages, colicins (Whelan, Sherburne & Taylor, 1997) and to other oxidative stress-generating antimicrobials (Taylor, 1999), which could be the result of transcriptional control by a common regulator, OxyR (Ni et al., 2014).

A number of reasons may explain the observed discordances among MIC values (i.e., Hg, Cu, As, etc.) and the respective resistance genes identified in this bacterium. For instance, P. glacincola BNF20 sensitivity to mercury could be a result of the absence of some genes (i.e., merT) belonging to the mer operon, which could render it non-functional (Boyd & Barkay, 2012). Similarly, the absence of the cusS gene (Cu sensor) in the P. glacincola BNF20 genome could be responsible for its copper sensitivity, in spite the presence of other genes that participate in Cu homeostasis (Rensing & Grass, 2003).

Tellurite resistance-associated ter genes are grouped in three different families: (i) TerC, encompassing transmembrane proteins, (ii) TerD, which includes the cytoplasmatic paralogs TerD, TerA, TerE, TerF and TerZ (Anantharaman, Iyer & Aravind, 2012), and (iii) TerB, representing proteins that are directly associated with the inner surface of the cell membrane, although they also have a cytoplasmatic localization (Alekhina, Valkovicova & Turna, 2011). As mentioned, TerC interacts with TerD, TerB and other proteins showing different cell functions (Turkovicova et al., 2016).

Most bacteria carrying ter genes display a similar transcriptional organization. Thus terZABCDEF, terZABCDE and TerABD present in E. coli O157:H7, Proteus sp. and D. radiodurans, respectively, are operons (Makarova et al., 2001; Toptchieva et al., 2003; Taylor et al., 2002). The Psychrobacter genus represents an exception to this rule, with terA lying in the opposite transcriptional orientation (Fig. S1).

Transcriptomic and proteomic assays have shown that terB is expressed when E. coli or D. radiodurans are exposed to tellurite (Anaganti et al., 2015; Taylor et al., 2002). TerB seems to be essential for tellurite resistance and interacts with some cytoplasmatic proteins such as the alpha subunit of ATP synthase, G subunit of the NADH-dependent quinone oxidoreductase and DnaK chaperone, among others (Alekhina, Valkovicova & Turna, 2011). Given that P. glacincola BNF20 lacks terB, we hypothesize that there must be another gene product that mediates tellurite resistance.

Based on their genetic background, ter genes have also been classified into different groups (I–IV) (Anantharaman, Iyer & Aravind, 2012). In this context and given its similitude with the ter genes found in Psychrobacter sp. PRwf-1, P. glacincola BNF20 would belong to group I, which contains a gene encoding a protein exhibiting the AIM24 domain, also found in the P. glacincola BNF20 TIGR00266 protein. Although no role has been ascribed to it in prokaryotes, in higher organisms it is an internal membrane protein related to mitochondrial biogenesis which is required for yeast respiration (Deckers et al., 2014). The AIM24 domain exhibits a double beta-helix folding, which is frequently found in genes neighboring TerD, suggesting that both proteins could interact (Anantharaman, Iyer & Aravind, 2012).

Deciphering the origins of bacterial operons is not straightforward, and there are some hypotheses that try to explain their formation. An interesting example is the piecewise model, which states that the his operon (hisGDCBHAFIE) was gradually formed. Phylogenetic analyses of the Proteobacterial phylum his genes showed their progressive grouping, which suggests that they were located in nearby zones of the chromosome in closely related microorganisms. Following, new events ended with the formation of the hisBHAF central core and the whole operon (Fani, Brilli & Liò, 2005). A future hypothesis to test is whether the ter operon has a similar evolutionary origin.

To evaluate the taxonomical distribution of ter genes in the Bacterial kingdom, the 5,398 genomes retrieved from the NCBI’s RefSeq bacterial database were screened. About 48.6% of them (2,623 genomes) were found to contain ter genes. While at the family level most (68.7%) harbored one ter gene (chiefly terC) and 15.6% exhibited five (including P. glacincola BNF20), at the class level the number of genomes exhibiting at least one ter gene was Gammaproteobacteria (379), Alphaproteobacteria (253) and Bacilli (247). Finally and regarding phyla, Proteobacteria, Actinobacteria and Firmicutes had 867, 854 and 361 genomes containing at least one ter gene, respectively (Fig. 4, Table S4).

Within the Proteobacteria phylum, most families had only one ter gene, while others up to 7 (Morganellaceae, Yersiniaceae), 6 (Chromatiacceae, Budviciaceae), 5 (Moraxellceae, Burkholderiaceae), 4 (Erythrobacteraceae), etc. (Fig. 4). In this context, it would be interesting to carry out phylogenetic analyses to understand the evolution of these ter genes and how the currently known terZABCDEF operon was formed (Taylor, 1999; Whelan, Sherburne & Taylor, 1997).

Finally, it was found that—in general—Gram-positive microorganisms contain more ter genes than Gram-negative bacteria (Table S4). This is interesting because it is generally accepted that they also show higher tellurite resistance (Taylor, 1999). For instance, Streptomyces and Bacillus genera comprise 137 and 65 genomes carrying up to 5–6 ter genes, respectively, suggesting that ter gene copy number could be related to the high resistance to tellurite observed in S. coelicolor and Geobacillus stearothermophilus (Moscoso et al., 1998; Sanssouci et al., 2011).

Conclusions

A new species of Antarctic bacteria exhibiting high tellurite resistance was isolated and identified as P. glacincola BNF20. Although within the genus the percent of sequence coverage is low, its genomic sequence is similar to other uncharacterized genomes and contains a large number of genes implicated in metal(loid) resistance, especially chromate and tellurite. The transcriptional orientation of tellurite resistance (ter) genes in P. glacincola BNF20 is different to that described in other microorganisms and most likely do not function as an operon. The wide distribution of ter genes in the bacterial world suggests that they play an important physiological role.

Supplemental Information

Table S1 Psychrobacter genome dataset used in this study

Click here for additional data file.

Table S2 Average nucleotide identity (ANI) and per cent of alignment fraction among the indicated Psychrobacter species

Click here for additional data file.

Table S3 Metal(loid) resistance genes in the indicated Psychrobacter species

Click here for additional data file.

Table S4 Per cent of bacterial genomes exhibiting ter genes

Click here for additional data file.

Figure S1 Genetic context of ter genes in the Psyc hrobacter genus

Click here for additional data file.

We would like to thank Fraunhöfer Foundation for carrying out the novo assembly and gene annotation, especially to Paz Tapia and Jorge Valdés. We also thank Natalia Valdés from Universidad de Santiago de Chile for helping with the bioinformatic analysis. ECN thanks the high-performance computing facility from The George Washington University, Colonialone, for providing data storage, support, and computing power for genomic analyses (http://colonialone.gwu.edu).

Additional Information and Declarations

Competing Interests

Author Contributions

Data Availability

The authors declare there are no competing interests.

Claudia Melissa Muñoz-Villagrán conceived and designed the experiments, performed the experiments, analyzed the data, prepared figures and/or tables, authored or reviewed drafts of the paper.

Katterinne N. Mendez performed the experiments, analyzed the data, prepared figures and/or tables, authored or reviewed drafts of the paper.

Fabian Cornejo and Maximiliano Figueroand performed the experiments, authored or reviewed drafts of the paper.

Agustina Undabarrena analyzed the data, prepared figures and/or tables, authored or reviewed drafts of the paper.

Eduardo Hugo Morales authored or reviewed drafts of the paper.

Mauricio Arenas-Salinas performed the experiments, prepared figures and/or tables, authored or reviewed drafts of the paper.

Felipe Alejandro Arenas contributed reagents/materials/analysis tools, prepared figures and/or tables, authored or reviewed drafts of the paper.

Eduardo Castro-Nallar conceived and designed the experiments, analyzed the data, contributed reagents/materials/analysis tools, prepared figures and/or tables, authored or reviewed drafts of the paper.

Claudio Christian Vásquez conceived and designed the experiments, analyzed the data, contributed reagents/materials/analysis tools, authored or reviewed drafts of the paper.

The following information was supplied regarding data availability:

The genome sequence described here is accessible via GenBank accession number NZ_LIQB00000000.1.

The raw reads and genome assembly are publicly available in NCBI under Bioproject (NCBI) accession number PRJNA293364.

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
