# Peer review of "Comparative genomic analysis of a new tellurite-resistant Psychrobacter strain isolated from the Antarctic Peninsula"

_PeerJ, doi:10.7717/peerj.4402_

## Round 0.1 · original submission · Minor Revisions

I would like to add a few minor comments:

L277 – Change expression “A battery of tests” to for example numerous tests;
L420 –Bacillus and Burkholderia are not only different genera they are different phyla. This is not good example. Correct.
L507 – percent should be one word.

Reviewer 1 ·

Basic reporting

no comments

Experimental design

no comments - The experimental design is valid. Research question is well defined and meaning. Methods are described with sufficient detail and information to replicate.

Validity of the findings

no comments - The results and findings are also valid. The manuscript highlights the diverse nature of the Psychrobacter genus, provide insights into potential mechanisms of metal resistance. The data is robust and conclusions are well stated.

Additional comments

The manuscript 21147 “Comparative genomic analysis of a new tellurite-resistant Psychrobacter strain isolated from the Antarctic Peninsula” the draft genome of a strain of Psychrobacter and comparative analysis with other diverse Psychrobacter strains. This high tellurite-resistant strains was identified as Psychrobacter glacincola and ~100 metal resistance genes were identified. Comparative analysis showed strain BNF20 was evolutionarily divergent from other Antarctic Psychrobacter isolates. In general, I think this manuscript merit to be published in PeerJ.
1. This manuscript has more than 30 pages, in my opinion, it should be abbreviated.
2. line 110-line 112, are there any references or data to support this phenomenon?
3. line 126, “Bacteria were isolated at 25 ℃”, I wonder why the authors isolate bacteria at 25 ℃ and not at lower temperatures (5 or 10 ℃), for it well-known that Antarctic is the coldest on Earth.
4. Results 3.1, in my opinion, it is not sufficient to identify the strain as Psychrobacter glacincola merely based on 16S rRNA gene sequence and fatty acid composition.
5. line 255-256, “(King George Island,-266 62.183183, -58.933355)” should be deleted.
6. line 295-296, please make it clear that rRNA genes are organized as cluster or operon.
7. line 363-366, “sp.” should not be italic.
8. line 420-421, do Bacillus and Burkholderia belong to the same genus?

·

Basic reporting

The authors reported the genome sequence of a new Antarctic bacterial species and provides a bunch of bioinformatics and experimental analysis that genetically and phenotypically typify/characterize such new species.


#General language issues#

The paper is well written although some points may deserve an additional editing, for example:

• Line 49 and 50: missing subject

‘Finally, investigating the presence and taxonomic distribution of ter genes in the NCBI´s 49 RefSeq bacterial database (5,398 genomes, as January 2017), revealed that 2,623 (48.59%) showed at least one ter gene.’
2,623 what? Authors should add ‘genome’ or ‘organism’ or ‘bacteria’ or similar.

Same in the following line:
“At the family level, most (68.7%) harbored one ter gene and 15.6% exhibited five (including P. glacincola BNF20).”
Most …? The subject is missing.


• Line 52:
“Overall, our results highlight the diverse nature of the Psychrobacter genus”
What do authors exactly mean with ‘diverse’? Here a rephrasing may help clarifying the general scope.

• Line 268-269:
“Previously, it was identified as P. inmobilis by partial 16S rRNA gene sequence analysis”

I would rather suggest: “Initially, it was erroneously identified/labeled as P. inmobilis based on partial 16S rRNA gene sequence analysis”

• Line 311-312:
“No genome exhibiting >96.5% ANI and 60% alignment fraction was found, suggestive of genomes belonging to the same species”

Here text is highly misleading.


#Tables and figures#

Caption in Table 2 is incomplete, it should be edited to include also reference to the third column.

“Table 2. Minimal inhibitory concentrations (mM) of different metal(loid)s for P. glacincola BNF20”

Authors could possibly add ‘compared to MIC for E. coli BW25113’ (or similar).

Experimental design

no comment

Validity of the findings

The author state that about 100 genes possibly conferring resistance have been found in the genome, however these results are not supported by experimental tests (except few cases). Could you please give some details? How do you explain the discordance?

Additional comments

I would suggest to clearly state that only 16S RNA is available for the strains used in the phylogenetic tree. Readers may in fact argue that those strains should be also used in ANI test.
This would also enforce the statement in the abstract ‘… the bacterium was identified as Psychrobacter glacincola BNF20, making it the first genome sequence reported for this species.’ which is not reported elsewhere.

---

## Round 0.2 · Minor Revisions

Dear Dr. Vásquez,

Thank you for resubmission of the revised manuscript. During examining the revised manuscript and rebuttal letter I mentioned that not all concernes raised by reviewers were acknowledged.

For example:

First example. Original L430: "A similar trend has been observed in other bacterial species. In fact, genomic differences have been observed among microorganisms belonging to the same species, as occurs in the Bacillus sp. and Burkholderia sp. (Jiménez et al., 2013; Losada et al., 2010; Varghese et al., 2015)."

Responce to reviwers: "L420 –Bacillus and Burkholderia are not only different genera they are different phyla. This is not good example. Correct.
RESPONSE: Done. We agree with the editor that the example was incorrect. We have deleted this in the revised version."

Revised L457: "A similar trend has been observed in other bacterial species. In fact, genomic differences have been observed among microorganisms belonging to the same species, as occurs in the Bacillus sp. and Burkholderia sp. (Jiménez et al., 2013; Losada et al., 2010; Varghese et al., 2015)."

Second example “2.- line 110-line 112, are there any references or data to support this phenomenon?
RESPONSE: Done. As per reviewer’s request we have added the corresponding reference to back up the statement.”
But I did not find any added reference there.

Third example: “3.- line 126, “Bacteria were isolated at 25 ℃”, I wonder why the authors isolate bacteria at 25 ℃ and not at lower temperatures (5 or 10 ℃), for it well-known that Antarctic is the coldest on Earth.
RESPONSE: We thank the reviewer for this observation. We failed to be explicit as to what exactly was done. We tested three temperatures, 4, 25, and 37 ºC, and under laboratory conditions the bacterium grew faster at 25 ºC. This is now reflected in the manuscript.”
But there is no explanation about choice of the temperature in the revised manuscript.

Please proofread and double-check to make sure that you completed all needed revisions in the manuscript. Please upload a copy of the manuscript in “track-changes-mode” for convenience.

---

## Round 0.3 · accepted · Accept

I have evaluated your manuscript and the reviewers’ reports, and the reports indicate that manuscript should be accepted for publication in its present form.

Reviewer 1 ·

Basic reporting

no comments

Experimental design

no comments

Validity of the findings

no comments

Additional comments

The revised MS peerJ-21147 has been revised according to the reviewer's comments, and the major comments have been answered clearly in " peerj-21147-Letter_to_the_Editor_05JAN2018.doc".

·

Basic reporting

no comment

Experimental design

no comment

Validity of the findings

no comment

Additional comments

The authors have addressed all the weakness present in the previous version of the manuscript